# First Report of Accumulation of Lyngbyatoxin-A in Edible Shellfish in Aotearoa New Zealand from Marine Benthic Cyanobacteria

**DOI:** 10.3390/toxins16120522

**Published:** 2024-12-03

**Authors:** Laura Biessy, Jonathan Puddick, Susanna A. Wood, Andrew I. Selwood, Megan Carbines, Kirsty F. Smith

**Affiliations:** 1Cawthron Institute, Molecular Algal Ecology, Nelson 7010, New Zealand; jonathan.puddick@cawthron.org.nz (J.P.); susie.wood@lincoln.ac.nz (S.A.W.); andy.selwood@cawthron.org.nz (A.I.S.); kirsty.smith@cawthron.org.nz (K.F.S.); 2Faculty of Environment, Lincoln University, Lincoln 7647, New Zealand; 3Auckland Council, Environmental Evaluation & Monitoring Unit, Auckland 1010, New Zealand; megan.carbines@aucklandcouncil.govt.nz

**Keywords:** bioaccumulation, cyanoHAB, degradation, food safety, harmful algal blooms, liquid chromatography–tandem mass spectrometry, marine cyanotoxins, public health risk, toxicity

## Abstract

This study reports the first documented accumulation of lyngbyatoxin-a (LTA), a cyanotoxin produced by marine benthic cyanobacteria, in edible shellfish in Aotearoa New Zealand. The study investigates two bloom events in 2022 and 2023 on Waiheke Island, where hundreds of tonnes of marine benthic cyanobacterial mats (mBCMs) washed ashore each summer. Genetic analysis identified the cyanobacterium responsible for the blooms as *Okeania* sp., a genus typically found in tropical marine ecosystems. Analysis by liquid chromatography–tandem mass spectrometry indicated that the cyanobacteria produced a potent dermatoxin, lyngbyatoxin-a (LTA), and that LTA had accumulated in marine snails, rock oysters and cockles collected near the mats. Snails contained the highest levels of LTA (up to 10,500 µg kg^−1^). The study also demonstrated that the LTA concentration was stable in composted mats for several months. The presence of LTA in edible species and its stability over time raise concerns about the potential health risks to humans consuming LTA-contaminated seafood. This underlines the need for further studies assessing the risks of human exposure to LTA through seafood consumption, particularly as climate change and eutrophication are expected to increase the frequency of mBCM blooms. The study highlights the need to develop public health risk management strategies for mBCMs.

## 1. Introduction

Microalgae are the foundation of all marine food webs, supporting marine ecosystems, fisheries and aquaculture, and they are a substantial carbon sink [1,2]. However, some species can form algal blooms—defined as the rapid growth and accumulation of microalgal biomass in aquatic ecosystems. Various types of microalgae can form harmful algal blooms (HABs), but the most common types are cyanobacteria, dinoflagellates, diatoms and haptophytes. HABs negatively impact many communities and can cause significant impacts on human and animal health due to the production of toxic or bioactive compounds [3]. They can also have major economic [4,5] and environmental repercussions [6]. Climate change has increased sea surface temperatures and altered ocean currents, expanding the habitable range of many HAB species into non-endemic regions [7,8].

Cyanobacterial HABs are most prevalent in freshwater environments, but in marine ecosystems, cyanobacteria can be found in planktonic and benthic habitats. On the benthos, they can form mats that become dominant on ocean floors and reefs [9,10,11]. The occurrence of extensive blooms of marine benthic cyanobacterial mats (BCMs) has been prominent in tropical and subtropical regions, but proliferations are increasing in temperate regions with climate warming and ongoing nutrient inputs [12,13,14]. BCMs can emit noxious odours when they bloom and decompose, producing volatile sulfur compounds, including hydrogen sulfide, methyl mercaptan and dimethyl sulfide [15]. Microbial activity during decomposition can consume dissolved oxygen, leading to lower water pH, hypoxia and fish mortality events [13,16].

Marine benthic cyanobacteria can produce toxins and other unknown secondary metabolites, with the most common toxin being lyngbyatoxin-a (LTA). LTA is mainly produced by species that were historically identified as *Lyngbya majuscula* but have now been reclassified into several genera, including *Lyngbya*, *Moorea*, *Okeania*, *Dapis* and others not yet determined [17,18]. LTA can cause acute dermatitis, commonly known as ‘swimmer’s itch’ or ‘seaweed dermatitis’, as well as ocular and respiratory irritation in humans due to aerosolised toxins [19,20]. Marine BCM species regularly wash up on the coastlines of Australia, causing dermatitis [21], and were suspected to be responsible for an irritant syndrome in Rarotonga of which 700 cases were reported and symptoms affected more than 30% of the coastal population of the island [22]. Toxic BCMs are an emerging health concern, and it is expected that their frequency and intensity of proliferation will increase due to climate change and anthropogenic eutrophication [9,23,24].

Recreational water activities may be a significant route of exposure to marine cyanotoxins. Potential routes of recreational exposure to marine cyanotoxins include direct contact via exposed parts of the body and cell material trapped under clothing (i.e., bathing suits or wetsuits), accidental swallowing of contaminated water and inhalation [25]. Occupational exposure may also occur through work directly in or on waterbodies affected by mats or scums [24]. Marine blooms of filamentous marine cyanobacterial species can dry on fishing nets, and contact with fresh and dried material has caused severe skin reactions as well as breathing difficulties for workers in the fishing industry [19,26].

Aside from these more common exposure routes, there have been reports of human mortalities and illnesses following accidental consumption of marine cyanotoxins. In the years 1993–1998 in Madagascar, consumption of meat from marine turtles led to poisoning episodes where 414 people were intoxicated, 29 of whom died (7% lethality) [27,28]. Described symptoms included acute gastritis, mouth ulcers, conjunctivitis, burning of the buccal mucosa and tongue with the appearance of papules, salivation, headache, weakness and fever [28]. The toxin implicated in these poisoning events from turtle meat was later identified as LTA [29], and it was deduced that the turtles had fed on seagrass that was harbouring mats of cyanobacteria of the genus *Lyngbya* [29]. Severe gastrointestinal illnesses and deaths have been linked to the consumption of seaweed contaminated with marine cyanobacteria and cyanotoxins, including LTA [17,30,31].

The minimum lethal dose (LD_100_) of LTA by intraperitoneal (ip) injection in mice has been estimated as 300 µg kg^−1^, and LTA was shown to induce severe damage to the villi’s capillaries, leading to bleeding in the small intestine; sublethal doses caused erosion in the stomach and the small and large intestines and inflammation in the lung [32]. Using topical application, the cutaneous median effective dose (ED_50_) was estimated to 4800 µg kg^−1^ [25]. No data on LTA-induced repeated toxicity are currently available, but LTA has been shown to have tumour-promoting activities via the protein kinase C activation pathway [33,34]. Although they cause negative ecological effects and are detrimental to human health, little is known about the biological and chemical diversity of marine BCMs and their toxins [35,36] and the risk that they pose to humans.

To our knowledge, there have been no studies investigating the accumulation of marine cyanotoxins in edible shellfish. The present study addresses two exceptional events where hundreds of tonnes of marine BCMs washed up on the beaches of a popular tourism destination in Aotearoa New Zealand (New Zealand) during the summers of 2022 and 2023. This is also in a region where wild and commercial shellfish are found and regularly harvested. Māori (New Zealand’s indigenous people) have a longstanding history and connection with the marine environment, with fish and aquatic invertebrates traditionally being their largest source of food [37]. Kaimoana (Māori: food gathered from the sea) has high cultural, economic and nutritional importance in New Zealand and many countries worldwide [38]. Therefore, it is important to understand whether toxins produced by BCMs accumulate in seafood species to assess the risk to human health. The aims of the present study were as follows: (1) to use genetic techniques to determine which species were responsible for these blooms; (2) to assess the levels of LTA in cyanobacterial mats, wild shellfish (oysters, cockles and sea snails) found near the mats. To achieve these aims, cyanobacterial mats and organisms were taken from several beaches during bloom events and analysed using liquid chromatography–tandem mass spectrometry (LC-MS/MS). A 4-month-long composting study was also undertaken to understand the stability and degradation of LTA in toxic cyanobacterial mats, which would further increase the understanding of the risks that LTA poses to human health over time, either by exposure or through the composting of removed mats.

The results from this study showed that shellfish, including gastropods, can accumulate marine cyanotoxins and that LTA present in mats does not degrade over time when composted for several months. This highlights that urgent research is needed to understand the risk this poses to human and animal health, in particular acute and chronic toxicity studies after consumption of seafood contaminated with LTA.

## 2. Results

### 2.1. Morphological and Genetic Characterization

Four strains of cyanobacteria were successfully isolated from Waiheke Island (Auckland)—two from Shelley Beach and two from Blackpool Beach. The strains had discoid cells arranged in trichomes covered by polysaccharide sheaths (Figure 1). The filament width ranged from 20 to 27 µm, the cell width from 20 to 25 µm and the cell length from 1.5 to 2 µm. The majority of the filaments appeared to rapidly outgrow their polysaccharide sheaths, leaving empty sheaths at the ends of the filaments (Figure 1B).

The partial 16S ribosomal RNA (rRNA) sequences (1369 base pairs (bp)) for all four strains were identical. The gene segment was compared to the BlastN database [39] to identify other, highly homologous sequences. The new ribosomal sequence had a 99% sequence homology match with other *Okeania* sp. sequences (99.13% identity with *Okeania plumata*, KC986934). A phylogenetic tree was constructed using 16S rRNA sequences from a variety of marine benthic cyanobacteria. Strains from Blackpool and Shelley Beaches grouped within the *Okeania* cluster. Although they formed their own clade within the *Okeania* group, it clustered most closely with *O. plumata* (Figure 2).

### 2.2. Lyngbyatoxin-A Analysis

LC-MS/MS analysis (Figure 3) revealed the presence of LTA in all BCM samples collected during two bloom events, with ‘wet weight’ concentrations ranging from 840 µg kg^−1^ (Sunkist Bay, February 2024) to 62,000 µg kg^−1^ (Shelley Beach, March 2023; Table 1). LTA was also detected in samples of whole bivalves (cockles, *Austrovenus stutchburyi*, and rock oysters, *Saccostrea glomerata*) and edible marine snails (*Lunella smaragda*) (Table 1), with the snails bioaccumulating the highest amount (up to 10,500 µg kg^−1^). The highest LTA level detected in cockles and oysters was approximately 50 µg kg^−1^. The compound malyngamide-S was also detected in all samples (concentrations are not presented because a standard was not available for calibration).

### 2.3. Degradation of Lyngbyatoxin-A in Cyanobacterial Mats Through Composting

Stability samples (t = 0) stored in a freezer and analysed over a 16-day period did not show any significant changes in LTA concentration (*p* = 0.31).

Concentrations of LTA were measured in the *Okeania* sp. mat composting samples collected from Surfdale Beach and Blackpool Beach (Figure 4A,B). The initial LTA concentration in the Blackpool Beach samples was ~2 mg kg^−1^ wet weight, lower than the concentrations observed in Surfdale Beach samples (~20 mg kg^−1^ wet weight). Overall, LTA concentrations did not decrease during the composting experiment in the mats from either beach. Linear regression analysis showed no statistically significant relationships for any of the composting buckets containing *Okeania* sp. mats collected from the two beaches (*p* = 0.11 for Surfdale Beach and *p* = 0.73 for Blackpool Beach; Figure 4C,D). The lack of a relationship between LTA concentrations and composting was also reflected in the weak coefficients of determination (*r*^2^ ≤ 0.51 for both beaches). The LTA concentrations observed in this portion of the composting study were also similar to the concentrations of the respective t = 0 samples, indicating that negligible change occurred during the composting experiment.

## 3. Discussion and Recommendations

The species responsible for the blooms of BCMs in New Zealand were from the genus *Okeania*. Of the *Okeania* species described to date, the strains were most similar to *Okeania plumata* morphologically [18], but genetic analysis indicates that the species has likely not yet been characterized. Further genetic characterization is currently under way to confirm the causative species (Biessy L. in prep). The taxonomy of marine cyanobacteria is challenging, and recent phylogenetic investigations have shown that tropical and subtropical marine cyanobacteria represent novel lineages, but many species have been wrongly assigned to existing genera based on existing classification systems, which are reliant on morphological similarities [35,36,40]. The genus *Okeania* was delineated from the genus *Lyngbya* in 2013, and this group is amongst the most abundant and broadly distributed in the tropical marine benthos [18,40]. This study reports the first detection of an *Okeania* species in temperate waters. Blooms of mat-forming cyanobacteria have been reported in New Zealand since the early 2000s in the same region and similar areas [41]. The species responsible was historically identified as *Lyngbya majuscula*, but genetic analysis was not performed at the time. Because of this, it is uncertain whether the species originally causing the blooms was the same as the current species. Little is known about the ecology of mat-forming cyanobacteria, and more research is needed to understand what physiochemical conditions promote blooms.

Tropical marine BCMs, including species of *Okeania*, have been the focus of biochemical research due to their prolific production of bioactive secondary metabolites [42]. Some of these bioactive compounds are potent toxins that can be hazardous to human health [9,17,19], in particular, the production of dermatoxic LTA. This study confirms the production of LTA by the *Okeania* species blooming in New Zealand, with concentrations in the mats reaching up to 62 mg kg^−1^. Residents living near the ‘hotspot’ beaches on Waiheke Island reported experiencing severe headaches and asthma lasting several months while the mats decomposed on the beaches (Waiheke Local Board, pers. comm.). The results of the LTA degradation study also demonstrated that LTA concentrations in the mats did not decrease during a 4-month compositing study and that no statistically significant negative relationships between LTA concentrations and composting time were observed. This shows that LTA was relatively stable during composting in sealed buckets, which was surprising given that other cyanotoxins are generally susceptible to bacterial degradation [43,44,45]. This highlights that benthic cyanobacterial mats containing LTA pose a long-term risk to human and environmental health once they wash up on shore and that precautions around these blooms should be taken at all times. It also means that the use of removed mats for food production compost is not advisable, increasing the costs for mats removal after they are cleared from the beach.

It is unclear how LTA and other secondary metabolites produced by BCMs affect aquatic and terrestrial organisms in the natural environment. It has been speculated that many of the bioactive metabolites produced by *Lyngbya* protect it from predation by crustaceans, herbivorous fish and gastropods [46]. However, some laboratory studies have shown that LTA can impact animal health. For example, ingestion of LTA in mice induced severe damage in the villi’s capillaries, leading to bleeding in the small intestine; erosion in the stomach, small intestine and large intestine and inflammation in the lung [32]. LTA is also considered a promoting agent for fibropapillomatosis in sea turtles, with these tumour-promoting agents shown to enhance viral synthesis and to reduce immune responses [47]. Waiheke Island is an important bird nesting ground in New Zealand [48], especially for shorebirds such as the threatened New Zealand dotterels (*Charadrius obscurus*, Tūturiwhatu; [49]) that are known to utilise the affected beaches for feeding and roosting. Bird deaths on Waiheke Island were reported around the same time as the blooms during the summer of 2023, but their stomach contents were not analysed for the presence of cyanobacterial mats or LTA. This should be an area of focus if blooms occur again. The cyanobacterial mats that wash up on the beach also potentially impact the birds and other terrestrial animals, mainly infauna, by modifying their feeding grounds either because they need to go further to find food or because their food source is removed when mats are removed from the beach.

Shellfish are an important food source for animals, including shorebirds, and humans. This study is the first to observe the accumulation of LTA in edible shellfish and gastropods. The marine snail *Lunella smaragda* (Gastropoda: Turbinidae) accumulated the highest amount of LTA (up to 10,500 µg kg^−1^). Marine snails have previously been shown to accumulate other marine biotoxins (e.g., saxitoxin, tetrodotoxin; [50,51,52]), and consumption of contaminated snails has led to human poisonings [53]. *Lunella smaragda* is a herbivorous grazing animal [54] and, it is likely that it was directly grazing on the toxic cyanobacterial mats. These snails are a traditional food for Māori and are regularly consumed around New Zealand [55]. Accumulation of LTA in these edible gastropods could lead to human poisoning through recreational harvesting and consumption. Cockles, *Austrovenus stutchburyi*, and rock oysters, *Saccostrea glomerata*, are also commonly harvested recreationally and consumed in New Zealand. This study showed that these filter-feeding shellfish can also accumulate LTA, but at lower concentrations than *L. smaragda* (reaching up to 50 and 65 µg kg^−1^, respectively, in this study). Evaluating the potential dose that shellfish and marine snail consumers might be exposed to, based on the current LTA concentration data, indicated that a dose of up to 70 µg kg^−1^ LTA could be consumed in a large serving of marine snails (400 g; Table 2). While this dose rate is less than the currently available toxicological information for LTA (ip LD_100_ = 300 µg kg^−1^, cutaneous ED_50_ = 4800 µg kg^−1^; [32,56]), the ip LD_100_ and cutaneous ED_50_ are not suitable reference doses for understanding human health risk from food consumption. It should also be noted that past examples of human poisonings and fatalities have been reported in Indo-Pacific areas following consumption of turtle meat or breastmilk contaminated with LTA [28,57,58]. To better establish the potential human health risk from LTA accumulation in seafood, more data on LTA concentrations in bivalves, marine snails and other seafood are required to build a more comprehensive database. Further knowledge of the negative health effects of LTA and a toxicological assessment of LTA are also required to define a robust reference dose for risk assessments.

This study highlights the potential risk marine benthic cyanobacterial mats and their toxins might pose to human, animal and ecosystem health. It is also worth noting that economically, blooms of BCMs can have significant impacts on both commercial fish catches and local authorities and communities through lost tourism and beach clean-up [60]. More research is needed, particularly more information on recreational exposure routes to LTA to inform a robust public health response (e.g., exposure levels via water or aerosols and how this relates to toxin concentrations in the mats). Future studies investigating exposure aspects would allow a more refined public health response, and the information will be reassuring for residents who live close to beaches affected by blooms of marine cyanobacteria [24,61]. This study is the first to analyse the impacts and accumulation of LTA in seafood, but more research is needed to understand the risks to human health, in particular acute and chronic toxicity studies following ingestion of LTA from contaminated seafood, and to understand the degradation of LTA following boiling or cooking of food. It is anticipated that the frequency and severity of marine cyanobacterial blooms will increase over the next few decades with rising eutrophication and sea temperatures [61,62]. Warming of water bodies has generally been shown to increase the growth rate of cyanobacteria to a greater extent than phytoplankton [63].

## 4. Materials and Methods

### 4.1. Sample Collection

Fresh black marine BCMs were collected from seven beaches on Waiheke Island and the Auckland region (New Zealand) between November and May during the summers of 2023 and 2024 (Figure 5). A survey of beaches around the entire island was undertaken in March 2023 to investigate how far the bloom had spread. In both summers, the majority of the bloom (over 400 tonnes each year, estimated by the local council during removal of the mats) washed up on three beaches: Blackpool, Surfdale and Shelley Beach (Figure 5 and Figure 6). These beaches are considered ‘hotspots’. Rock oysters (*Saccostrea glomerata*), cockles (*Austrovenus stutchburyi*; tuangi) and marine snails (*Lunella smaragda*; cat’s eyes, pūpū) were collected from the three ‘hotspot’ beaches in March 2023 and January 2024 (Figure 6). The mat and animal samples were chilled (under 10 °C) for up to 24 h before being processed in the laboratory. Subsamples of the mats were washed in sterile seawater and processed for further strain isolation. The remaining mat subsamples and the animal samples were frozen (−20 °C) upon arrival for toxin analysis.

### 4.2. Isolation of Cyanobacterial Strains

Cyanobacterial strains of the dominant species in BCMs collected from two beaches were washed with modified L1 media (Berthold D. in prep., modified from [64]), and individual strains were isolated in 12-well plates. After two months, single filaments were isolated by micro-pipetting and transferred to 50 mL plastic pottles (Biolab, New Zealand) containing modified L1 medium (approx. 30 mL). Samples were incubated under standard conditions (100 ± 20 mmol photons m^−2^ s^−1^; 12:12 h light/dark; 18 ± 1 °C) in the Cawthron Institute Culture Collection of Microalgae (isolated strain code: CAWD450; [65]). Morphological analyses of the benthic mat material and isolated strains were undertaken using an Olympus light microscope (BX41, Olympus, Wellington, New Zealand) paired with cellSens imaging software (Olympus, Standard version, NZ). Filament width and cell width and length were measured for 25 cells.

Four subsamples, targeting filaments, were taken from densely growing cultures and were individually placed in a tube containing bashing beads from a DNA extraction kit (DNeasy Powersoil Pro kit, Qiagen, Germantown, MD, USA). The DNA was then extracted following the manufacturer’s instructions using an automated homogenizer (1600 MiniG Automated Tissue Homogenizer and Cell Lyser, SPEX SamplePrep, Metuchen, NJ, USA) and a robotic workstation for DNA extraction (QIAcube, Qiagen). The 16S rRNA gene (~1 kb) and 16S-23S rRNA internal transcribed spacer (ITS) region were amplified using the primer pairs 359F/1487R and 1337F/23S30R, respectively [66,67]. Thermal cycling conditions were 95 °C for 5 min followed by 95 °C for 30 s, 52 °C for 45 s and 72 °C for 45 s for 35 cycles, followed by a final extension of 72 °C for 5 min. Amplification products were purified (AxyPrep PCR clean-up kits, Axygen, Union City, CA, USA) and directly sequenced (in both directions using PCR primers) by an external contractor (Genetic Analysis Services, University of Otago, Dunedin). Forward and reverse sequences were aligned using Geneious software (version 2024.0) and conflicts resolved by manual inspection. Sequences obtained were aligned using the Clustal Omega algorithm [68] with publicly available selected cyanobacteria sequences from GenBank (www.ncbi.nlm.nih.gov). Phylogenetic analysis was undertaken using the Geneious Tree Builder method based on the Jukes–Cantor distance model, with the cyanobacterial species *Microcystis aeruginosa* selected as the outgroup. Sequences generated during this work were deposited in the NCBI GenBank database under submission number SUB14717735.

### 4.3. Lyngbyatoxin-A Analysis

Whole bivalves, specifically rock oysters (*S. glomerata*) and cockles (*A. stutchburyi*), were removed from the shell, and multiple specimens were finely diced into a composite sample (all organs combined). Marine snail flesh (*L. smaragda*) was extracted from the shell either using tweezers or by crushing the shell and separating the flesh from the pieces of broken shell. Multiple specimens were finely diced into a composite sample.

Mat and tissue composite samples were weighed (2 g ± 0.02 g) into a 50 mL tube, and freshly made 90% methanol and 0.1% formic acid solution (18 mL) was added to each tube. For the tissue samples, an extra step was undertaken to homogenise the tissue (Ultra-Turrax^®^, IKA^®^, Wilmington, NC, USA) for 1 min. Tubes were sonicated in an ice bath for 30 min and clarified by centrifugation (3200× *g* for 10 min), and 1 mL of the clarified supernatant was transferred to a LC vial and stored at −20 °C until further LC-MS/MS analysis.

LTA concentrations were determined by LC-MS/MS, following the method described by Videau et al. [69], where compounds were separated on an Acquity I-Class ultra-performance liquid chromatography system (Waters Co.) using a C_8_ column (Waters Acquity BEH-C_8_, 1.7 µm, 50 × 2.1 mm) and a gradient of milli-Q water + 0.1% formic acid (Solvent A) to acetonitrile + 0.1% formic acid (Solvent B). Sample extracts were loaded at 60% Solvent B before separation with a linear gradient to 90% Solvent B over 2.5 min. The column was flushed with 90% Solvent B for 1 min before returning to 60% Solvent B to equilibrate for 1.4 min.

Sample components were analysed on a Xevo-TQS mass spectrometer (Waters Co., Auckland, NZ, USA) operated in positive-ion electrospray ionisation mode (capillary voltage 0.45 kV; cone voltage 30 V; nitrogen desolvation gas 200 L h^−1^ (200 °C); cone gas 150 L h^−1^; nebuliser pressure 2.5 bar) with multiple reaction monitoring channels for LTA (*m/z* 438.3 > 410.3, quan; *m*/*z* 438.3 > 393.3, qual) and malyngamide-S (*m*/*z* 484.3 > 256.2, quan; *m*/*z* 484.3 > 434.3, qual). An external calibration curve of lyngbyatoxin-a (Cayman Chemical, MI, USA) was prepared in 80% acetonitrile + 0.1% formic acid (1–20 ng/mL). The limit of detection (LoD) for the analytical method was 0.01 ng/mL, and the limit of quantitation (LoQ) was 0.05 ng/mL. This equates to an LoD of 0.1 µg/kg and an LoQ of 0.5 µg/kg in the samples (using the sample preparation methodology described here). A standard for malyngamide-S was not available; therefore, only the presence/absence of this compound was noted.

### 4.4. Degradation of Lyngbyatoxin-A in Cyanobacterial Mats Through Composting

Benthic cyanobacterial mats from two beaches (Blackpool and Surfdale) were collected in March 2023 and transported within 24 h to a holding lot. Mats from each beach were divided between two sealable buckets (*n* = 4 four buckets total) and mixed. Initial samples (t = 0) were collected in 50 mL plastic tubes from each bucket and sent to the laboratory. The buckets were then sampled periodically seven times over approximately 4 months (127 days, Table 3). At each sampling point, the contents of the buckets were mixed, and four samples from each bucket were collected in 50 mL plastic tubes. Tubes were stored frozen until they were couriered to the laboratory for further extraction and analysis.

Sample tubes were defrosted, and 2 g subsamples were prepared as described in Section 4.4. Extracts were diluted with 80% acetonitrile + 0.1% formic acid directly in glass autosampler vials (dilutions were undertaken as 1/10 for the Blackpool samples and 1/50 for the Surfdale samples). LTA concentrations were determined by LC-MS/MS as described in Section 4.4.

To verify that freezing samples until analysis did not result in LTA degradation, initial samples (t = 0, *n* = 4 for each beach) were collected and analysed immediately. Replicate subsamples were weighed out and stored in a freezer. With each subsequent analysis performed over the following month, replicates from each sample were extracted and analysed (*n* = 4 per run), representing a 16-day stability period when stored in a freezer—sufficient time for sample transportation and analysis.

### 4.5. Biosafety Procedure and Sampling

Mats were collected under Cawthron Institute MPI special permit 822. Appropriate PPE (personal protective equipment, e.g., gloves and face masks) was used, when appropriate, during sampling and analysis to avoid contact with potential dermatoxins in the samples. After processing, toxic materials were placed in biohazard bags and autoclaved before being disposed of. Extracted toxic material has been kept frozen for potential further analysis.

## Figures and Tables

**Figure 1 toxins-16-00522-f001:**
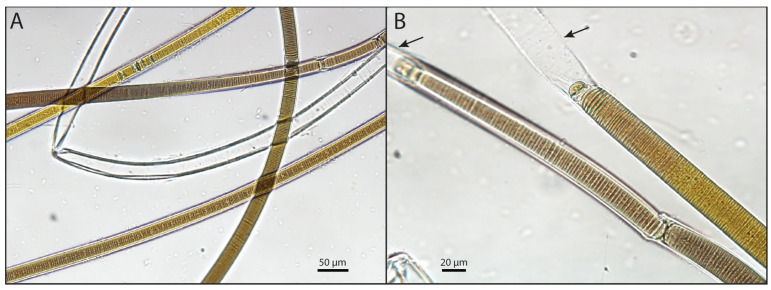
Light micrographs of *Okeania* sp. filaments isolated from Waiheke Island, New Zealand. (**A**) Trichomes are enveloped in sheaths to which epiphytic bacteria are attached. (**B**) Photograph showing the empty sheaths at the end of the filaments (black arrows).

**Figure 2 toxins-16-00522-f002:**
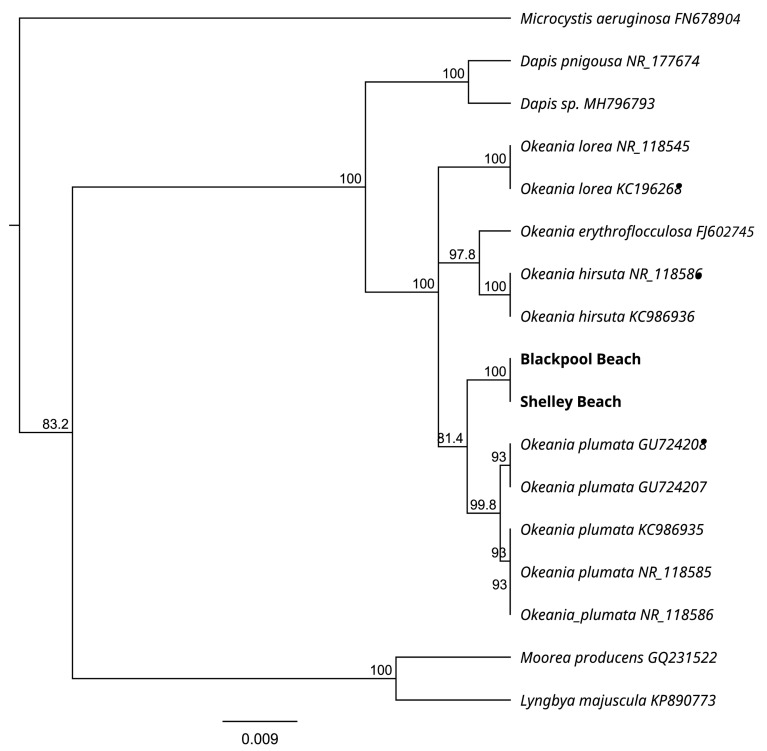
Phylogenetic tree based on 16S ribosomal RNA (rRNA) gene sequences of the Waiheke Island (New Zealand) cyanobacterial mats. The tree was based on the Jukes–Cantor model, using the UPGMA tree-building method. The percentage of trees in which the associated taxa clustered together is shown next to the branches. GenBank submission number: SUB14717735.

**Figure 3 toxins-16-00522-f003:**
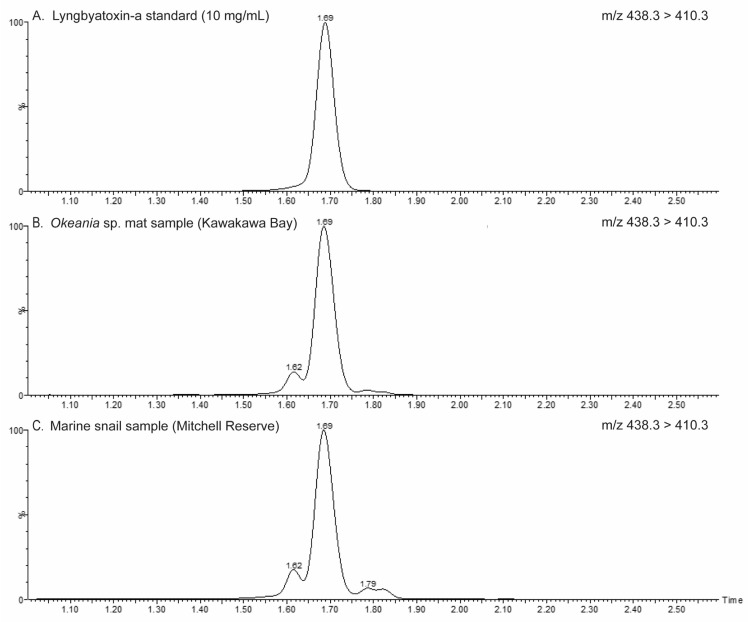
Total ion chromatograms of the multiple reaction monitoring channel analysing for (**A**) lyngbyatoxin-a in a standard and extracts from (**B**) an *Okeania* sp. mat and (**C**) a marine snail (*Lunella smaragda*) collected from the vicinity of *Okeania* sp. mats.

**Figure 4 toxins-16-00522-f004:**
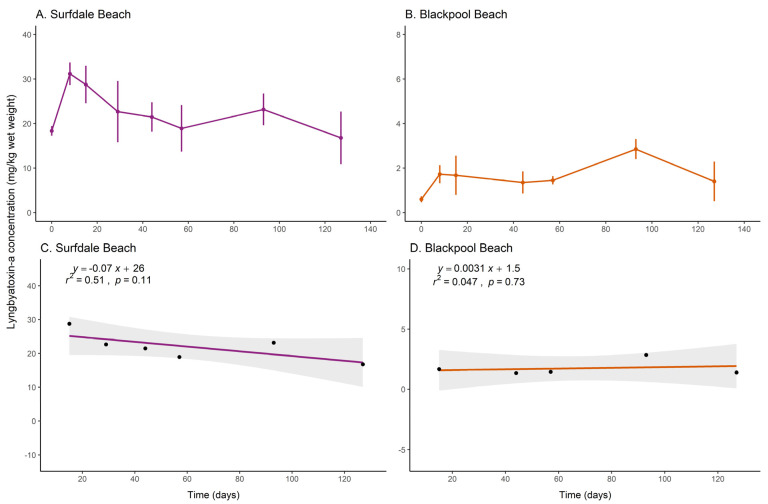
Mean concentrations of lyngbyatoxin-a (LTA) for the composting samples collected from (**A**) Surfdale Beach and (**B**) Blackpool Beach. The error bars represent the standard error for four replicate subsamples. (**C**,**D**) Linear regression analysis of mean LTA concentrations in *Okeania* sp. mat composting samples from 8 to 127 days (t = 1 to t = 7; the shaded region indicates the 95% confidence interval).

**Figure 5 toxins-16-00522-f005:**
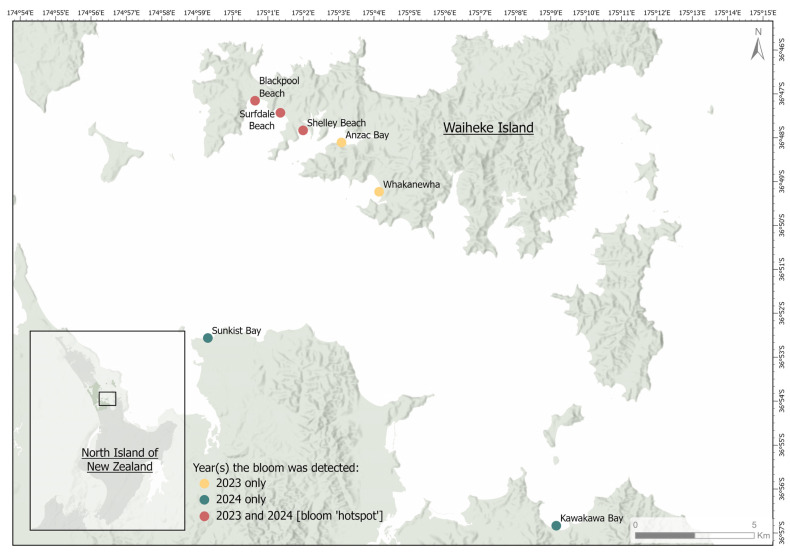
Locations of benthic cyanobacterial mat collection sites from Waiheke Island and the Auckland region (Aotearoa New Zealand).

**Figure 6 toxins-16-00522-f006:**
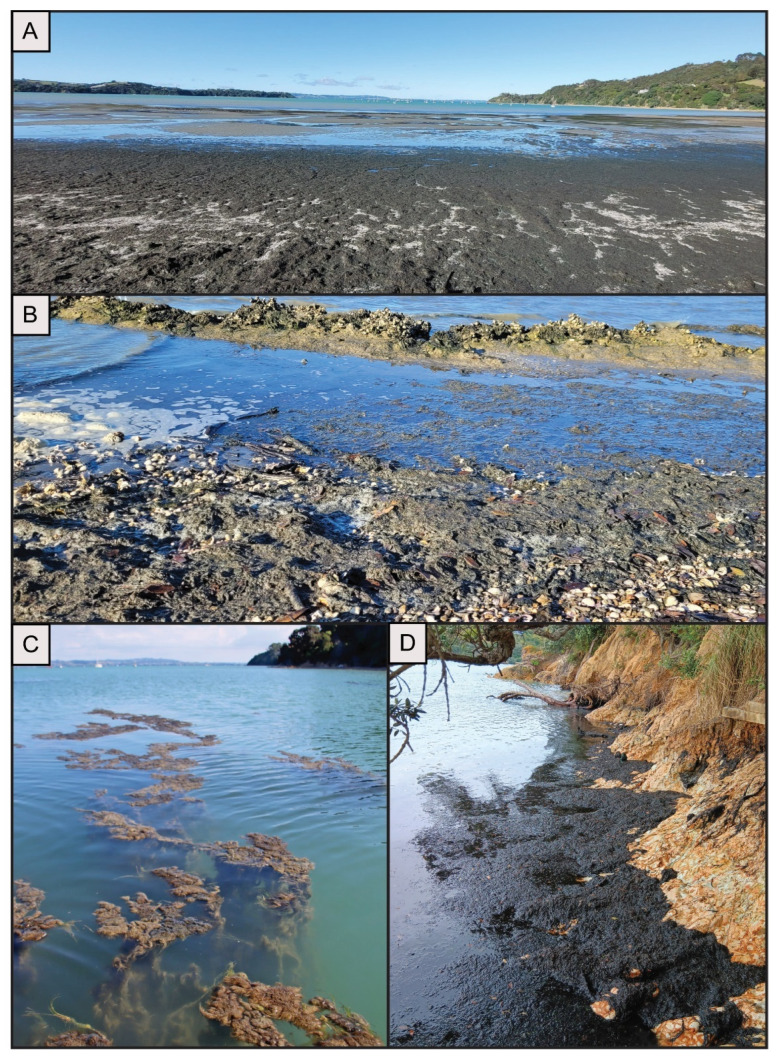
Images from Waiheke Island. (**A**) Tonnes of benthic marine cyanobacterial mats (BCMs) washed up on Blackpool Beach 2023 in March; (**B**) BCMs washing up on shellfish beds on Shelley Beach in March 2023; (**C**) BCMs floating to the surface near Blackpool Beach in January 2024 (credit: Merrie Hewetson); and (**D**) washed-up BCMs on Surfdale Beach in December 2023 (credit: Auckland Council).

**Table 1 toxins-16-00522-t001:** Lyngbyatoxin-a concentrations in different benthic cyanobacterial mat and shellfish samples (cockles, *Austrovenus*; rock oysters, *Saccostrea glomerata*; snails, *Lunella smaragda*) collected during bloom events on Waiheke Island, New Zealand.

Sites	Beaches	Time Sampled	Sample Type	Lyngbyatoxin-A Concentration (µg kg^−1^ Wet Weight)
Shelley Beach	North Beach	Mar-23	Mats	62,000
	Middle Beach	Mar-23	Oysters ^a^	35
			Cockles ^a^	16
			Snails	154
	South Beach	Mar-23	Oysters	51
			Mats	12,300
	North Beach	May-23	Mats	18,900
Blackpool Beach		Mar-23	Mats	32,400
			Cockles	50
		Dec-23	Mat ^a^	30,000
		Jan-24	Snails	2.1
			Cockles	2.8
Surfdale Beach	Mitchell Reserve	Dec-23	Mats ^a^	20,000
		Jan-24	Snails	10,500
			Cockles	2.6
	Surfdale North	Jan-24	Snails	3.2
		Feb-24	Mats	2.2
Sunkist Bay		Feb-24	Mats	840
Kawakawa Bay		Feb-24	Mats	16,700

^a^ The average of two data points is presented.

**Table 2 toxins-16-00522-t002:** Potential lyngbyatoxin-a (LTA) dose rates from seafood consumption based on the current dataset.

Species	Maximum LTA Conc (µg kg^−1^; Wet Weight) ^a^	Serving Size (g) ^b^	Dose per Serving (µg)	Dose Rate for 60 kg Adult (µg kg^−1^)
Oyster	65	250	16	0.3
		400	26	0.4
Cockle	50	250	13	0.2
		400	20	0.3
Snail	10,500	250	2625	44
		400	4200	70

^a^ The maximum value was used due to the small datasets available for each species (*n* = 3 to *n* = 5). ^b^ Shellfish serving sizes are discussed in Finch, et al. [59]: 250 g represents a serving size that would cover 97.5% of consumers in most countries where data are available, and 400 g represents the 95th percentile of shellfish consumption in Germany and the Netherlands.

**Table 3 toxins-16-00522-t003:** Sampling time points for the benthic cyanobacterial mat composting study.

Sampling Date	Sample	Number of Days Composting
26 March 2024	Stability Samples	0
26 March 2024	t = 0	0
3 April 2024	t = 1	8
10 April 2024	t = 2	15
24 April 2024	t = 3	29
9 May 2024	t = 4	44
22 May 2024	t = 5	57
27 June 2024	t = 6	93
31 July 2024	t = 7	127

## Data Availability

The data presented in this study are openly available in GenBank submission number: SUB14717735. [GenBank] [https://www.ncbi.nlm.nih.gov/nuccore/PQ308398.1, accessed on 28 October 2024] [PQ308398.1].

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
