# Peer review of "First Report of Accumulation of Lyngbyatoxin-A in Edible Shellfish in Aotearoa New Zealand from Marine Benthic Cyanobacteria"

_toxins, 2024, doi:10.3390/toxins16120522_

Round 1
Reviewer 1 Report
Comments and Suggestions for Authors
The authors examined LTA in cyanobacteria and animals that potentially consumed cyanobacteria. The project is well designed and executed. A few minor issues need to be addressed.
1. Have the authors tried to examine the presence of cyanobacterial cells in the guts and other tissues of the animals?
2. The authors should elaborate about the tissues analyzed for LTA toxin.
line 139-142: Which tissues of each animal were analyzed for LTA?
line179-180: This sentence should be elaborated to include more details of species identification. These strains were similar to Okeania plumata in terms of mophological features, or molecular sequences?
Author Response
Reviewer 1
The authors examined LTA in cyanobacteria and animals that potentially consumed cyanobacteria. The project is well designed and executed. A few minor issues need to be addressed.
The authors would like to thank the reviewer for their positive review and comments. The minor issues have been amended in the manuscript as requested.
- Have the authors tried to examine the presence of cyanobacterial cells in the guts and other tissues of the animals?
This is a great question. Unfortunately, we could not look at the presence of cyanobacterial cells in the guts as the animals were frozen upon arrival, making microscopical observation impossible. If the bloom was to happen again, we will look into this further.
- The authors should elaborate about the tissues analyzed for LTA toxin.
line 139-142: Which tissues of each animal were analyzed for LTA?
Whole shellfish were analysed, not specific tissues.
Details were added to line 139 (“whole bivalve” was specified) and in the methods section (lines 342-344; the sentence now reads “Whole bivalves, rock oysters S. glomerata and cockles A. stutchburyi, were removed from the shell and multiple specimen were finely diced into a composite sample (all organs combined)”).
line179-180: This sentence should be elaborated to include more details of species identification. These strains were similar to Okeania plumata in terms of morphological features, or molecular sequences?
Details were added as requested to lines 179-182 (in the revised manuscript). The sentence now reads; “The species responsible for the blooms of BCMs in New Zealand were from the genus Okeania. Of the Okeania species described to date, the strains were most similar to Okeania plumata morphologically [18], but genetic analysis indicates that the species has likely not yet been characterised.”
Reviewer 2 Report
Comments and Suggestions for Authors
The occurrence of toxic marine benthic cyanobacterial mats is an emerging threat in many coastal regions. This study reports the contamination of seafood (bivalves and gastropods) with Lyngbyatoxin-a and provides evidences that cyanobacterium species responsible for the blooms and toxin-producing species is a cyanobacteria from the Okeania genus.
The manuscript is generally well written, the methods are adequate and the results and discussion section provide advance on the scientific knowledge. I can recommend this manuscript for publication after some minor corrections:
1- Line 138-139: These concentrations refers to BMC fresh weight? Please state.
2- Line 147: Fig 3. In the figure captions indicate the m/z transition used. I also suggest to add the signal of both transitions, qualitative and quantitative. To include the mass spectra of the LTA detected in mats would be interesting.
3- Line 297, Figure 5. Please include Lat/Long coordinates.
4- Line 316: Section 4.2 and 4.3 have the same subtitle, please revise.
5- Line 344: What is the meaning of fresh 90% MeOH? I suggest delete “fresh”.
6- Line 349: Is the LCMSMS determination method based on any method previously developed? If so, please state. If not please indicate more details, such as method recovery, LOD & LOQ, MS source data…
Author Response
Reviewer 2
The occurrence of toxic marine benthic cyanobacterial mats is an emerging threat in many coastal regions. This study reports the contamination of seafood (bivalves and gastropods) with Lyngbyatoxin-a and provides evidences that cyanobacterium species responsible for the blooms and toxin-producing species is a cyanobacteria from the Okeania genus. The manuscript is generally well written, the methods are adequate and the results and discussion section provide advance on the scientific knowledge.
The authors would like to thank the reviewer for their positive comments. Minor corrections were made throughout the manuscript as requested.
I can recommend this manuscript for publication after some minor corrections:
1- Line 138-139: These concentrations refers to BMC fresh weight? Please state.
We have clarified in the sentence suggested and Tables 1 and 2 that concentrations were by ‘wet weight’.
2- Line 147: Fig 3. In the figure captions indicate the m/z transition used. I also suggest to add the signal of both transitions, qualitative and quantitative. To include the mass spectra of the LTA detected in mats would be interesting.
As suggested by the reviewer, we have added the mass transition information to Figure 3.
Unfortunately, multiple-reaction monitoring data does not collect qualitative mass spectra, just targeted data that is displayed as a chromatogram (as presented in Figure 3). No changes were made to the manuscript in relation to the second part of this comment.
3- Line 297, Figure 5. Please include Lat/Long coordinates.
Coordinates were added to the map as requested.
4- Line 316: Section 4.2 and 4.3 have the same subtitle, please revise.
We thank the reviewer for noticing. This has now been modified.
5- Line 344: What is the meaning of fresh 90% MeOH? I suggest delete “fresh”.
Modified as “freshly made” as requested.
6- Line 349: Is the LCMSMS determination method based on any method previously developed? If so, please state. If not please indicate more details, such as method recovery, LOD & LOQ, MS source data…
We added information on the study the LC-MS/MS method was based on (line 353-354 in the revised manuscript), more details on the LC-MS/MS method were added (lines 362-364 in the revised manuscript), as well as information on the LoD and LoQ (line 367-370 in the revised manuscript).
Reviewer 3 Report
Comments and Suggestions for Authors
Dear Editor Greetings,
I have read in detail the article entitled “First report of accumulation of lyngbyatoxin-a in edible shellfish in Aotearoa New Zealand from marine benthic cyanobacteria”.
The article is very interesting regarding the first documented accumulation of lyngbyatoxin-a (LTA), a cyanotoxin produced by marine benthic cyanobacteria, in edible shellfish in Aotearoa New Zealand.
I would also request that you include the biosafety procedure for disposal of the toxic materials, as well as what sanitary authorization was used for sample collection. It would be important to specify if the tissues analyzed corresponded only to the consumables, and if there was any estimation or temporal evaluation of the duration of the toxins in the tissues of the different species analyzed.
Comments:
Line 27-28: Please include a more acceptable definition for HABs.
Fig1B: Please include in the image an arrow to be more descriptive to what you wish to highlight.
Line 141: Could you please link your bioaccumulation data to the food web?
Fig 3: Please continue with the standard description you have used in previous figures (a,b,c).
Fig 5: Please include a better quality figure (tone, contrast).
Line 280: What was the procedure used for counting cyanobacteria?
Line 281: When did you collect individuals per species? Consider that the interaction with HABs is different between species, so some are exposed by filtration and others by predation. Another point, why don't you use the term cyanoHABs directly?.
Line 337: Please include the LOD and LOQ of the technique used.
Was the matrix effect evaluated?
Author Response
Reviewer 3
Dear Editor Greetings,
I have read in detail the article entitled “First report of accumulation of lyngbyatoxin-a in edible shellfish in Aotearoa New Zealand from marine benthic cyanobacteria”. The article is very interesting regarding the first documented accumulation of lyngbyatoxin-a (LTA), a cyanotoxin produced by marine benthic cyanobacteria, in edible shellfish in Aotearoa New Zealand.
The authors would like to thank the reviewer for their detailed review. Changes were made as requested throughout the manuscript and some explanations are below:
I would also request that you include the biosafety procedure for disposal of the toxic materials, as well as what sanitary authorization was used for sample collection. It would be important to specify if the tissues analyzed corresponded only to the consumables, and if there was any estimation or temporal evaluation of the duration of the toxins in the tissues of the different species analyzed.
A new Section 4.5 – “Biosafety procedure and sampling” (lines 405-409 in the revised manuscript) has been added to the manuscript describing the sampling and biosafety procedure as requested. This reads; “Mats were collected under the Cawthron Institute MPI special permit 822. Appropriate PPE (personal protective equipment; e.g., gloves, face masks) were used, when appropriate, during sampling and analysis to avoid contact with potential dermatoxins in the samples. After processing, toxic materials were placed in biohazard bags and autoclaved before being disposed of. Extracted toxic material has been kept frozen for potential further analysis.”
Details were added line 139 and in the methods section, (lines 339-342) to mention that whole shellfish were analysed, not specific tissues. Shellfish that were analysed are typically consumed whole, making no difference to which organs accumulated the toxin. Dates of collection for mats and shellfish are mentioned in Table 1 – this shows that the different shellfish species accumulated the toxin over a long period of time.
Comments:
Line 27-28: Please include a more acceptable definition for HABs.
Thank you for pointing this out, the sentence has now been restructured to be more accurate. The sentences now read (lines 25-30 in the revised manuscript); “Microalgae are the foundation of all marine food webs, they support marine ecosystems, fisheries and aquaculture, and are a substantial carbon sink [1,2]. However, some species can form algal blooms – defined as the rapid growth and accumulation of microalgal biomass in aquatic ecosystems. Various types of microalgae can form harmful algal blooms (HABs), but the most common types are cyanobacteria, dinoflagellates, diatoms and haptophytes.”
Fig1B: Please include in the image an arrow to be more descriptive to what you wish to highlight.
Arrows were added to Fig 1B as requested.
Line 141: Could you please link your bioaccumulation data to the food web?
A possible reason for higher bioaccumulation in the marine snails has been mentioned in the discussion (lines 238 – 240).
Fig 3: Please continue with the standard description you have used in previous figures (a,b,c).
Figure 3 was modified as requested.
Fig 5: Please include a better quality figure (tone, contrast).
The quality of the map is quite high and there must have been an issue with the reviewer’s version from the upload of the manuscript. The file has now been saved as a TIFF with higher quality, we hope this helps in the reviewer’s version.
Line 280: What was the procedure used for counting cyanobacteria?
The Auckland Council (from Megan Carbines, co-author on the paper) removed the mats from the beaches and a barge had to be used for the removal, making it possible to estimate the biomass that had washed up on the beach. Details from the source were added to the manuscript.
Line 281: When did you collect individuals per species? Consider that the interaction with HABs is different between species, so some are exposed by filtration and others by predation. Another point, why don't you use the term cyanoHABs directly?.
The months of the collection points are stated in section 4.1 for all samples.
We agree that the interaction with HABs is different between species, we have described this in the paragraph lines 234-247, where we explained the difference between a herbivorous grazing snail versus filter feeding bivalves.
We did not use the term cyanoHAB in the manuscript but we have now added it to the keywords.
Line 337: Please include the LOD and LOQ of the technique used.
We have now added this detail as suggested (lines 367-370 in the revised manuscript); “The limit of detection (LoD) for the analytical method was 0.01 ng/mL and the limit of quantitation (LoQ) was 0.05 ng/mL. This equates to an LoD of 0.1 µg/kg and an LoQ of 0.5 µg/kg in the samples (using the sample preparation methodology described here).”
Was the matrix effect evaluated?
Matrix effects weren’t evaluated during this study, but we plan to undertake validation work on the analytical method in the future.